# Use of Infrared Thermography in Medical Diagnosis, Screening, and Disease Monitoring: A Scoping Review

**DOI:** 10.3390/medicina59122139

**Published:** 2023-12-09

**Authors:** Dorothea Kesztyüs, Sabrina Brucher, Carolyn Wilson, Tibor Kesztyüs

**Affiliations:** 1Medical Data Integration Centre, Department of Medical Informatics, University Medical Centre, Georg-August University Göttingen, 37073 Göttingen, Germany; carolyn.wilson@med.uni-goettingen.de (C.W.); tibor.kesztyues@med.uni-goettingen.de (T.K.); 2Institute for Distance Learning, Technical University of Applied Sciences, 13353 Berlin, Germany

**Keywords:** thermography, diagnostic imaging, scoping review

## Abstract

Thermography provides non-invasive, radiation-free diagnostic imaging. Despite the extensive literature on medical thermography, a comprehensive overview of current applications is lacking. Hence, the aim of this scoping review is to identify the medical applications of passive infrared thermography and to catalogue the technical and environmental modalities. The diagnostic performance of thermography and the existence of specific reference data are evaluated, and research gaps and future tasks identified. The entire review process followed the Joanna Briggs Institute (JBI) approach and the results are reported according to PRISMA-ScR guidelines. The scoping review protocol is registered at the Open Science Framework (OSF). PubMed, CENTRAL, Embase, Web of Science, OpenGrey, OSF, and PROSPERO were searched using pretested search strategies based on the Population, Concept, Context (PCC) approach. According to the eligibility criteria, references were screened by two researchers independently. Seventy-two research articles were identified describing screening, diagnostic, or monitoring studies investigating the potential of thermography in a total of 17,314 participants within 38 different health conditions across 13 therapeutic areas. The use of several camera models from various manufacturers is described. These and other facts and figures are compiled and presented in a detailed, descriptive tabular and visual format. Thermography offers promising diagnostic capabilities, alone or in addition to conventional methods.

## 1. Introduction

The control of body temperature has been an important health indicator for clinical diagnosis and treatment since the time of Hippocrates, and it is said that Hippocrates applied wet mud to the skin to observe the faster drying of a tumorous swelling [1]. The skin temperature in a healthy person under standardized environmental conditions stays within a narrowly defined range and depends on the blood circulation in the lower layers of the skin. Therefore, deviations in surface temperature can have many causes, such as inflammation, malignancy, and infection [2]. Medical infrared thermography (IRT) offers a fast, painless, non-contact, non-invasive, and radiation-free method of photographically imaging temperature differences of the skin surface, allowing arbitrary repetitions of recording [3]. Thermal imaging of the body surface can detect minute changes in the underlying tissues, which indicates neurological and vascular, as well as metabolic pathologies. For this purpose, modern infrared thermal imaging cameras are used, which have a high temperature sensitivity and produce high-resolution thermal images. Thermographic devices detect the infrared radiation emitted by the body surface, which is invisible to humans, convert it into electrical signals, and a thermogram can be created, which shows temperatures pictorially in different colours or gray tones [4]. Infrared thermography is divided into several application modalities: active thermography, following a thermal stimulation; dynamic thermography, the recording of the temperature over time; and passive thermography, without stimulation [5]. Since an investigation of all methods is beyond the scope of a reportable review, the present investigation is limited to passive thermography.

Today, a broader spectrum of different imaging modalities is available, from ultrasound to X-ray technologies to magnetic resonance imaging, most of them relying on contrast agents and/or radiation [6]. In contrast, thermal imaging is a radiation-free and predominantly passive technique that detects infrared radiation without the use of contrast agents, but it has so far only partially established itself as a standard procedure in routine medical diagnostics. In some application areas, such as non-contact body temperature measurement, thermography is already state of the art and has only recently proven its practicality in mass use for temperature measurements during the COVID-19 pandemic [7]. In many other areas, it is still experimental or considered an outsider method, but the increasing number of published studies and reviews indicate a great interest among medical researchers [8]. Technological progress, both in thermographic devices and in potential subsequent image processing and evaluation by artificial intelligence, opens up great opportunities for a wide range of applications of this non-invasive methodology in the medical-diagnostic field. A comprehensive, evidence-based overview of the current state of knowledge and development is still missing for a breakthrough with regard to a broad application in the entire medical field. This is essential for the further development and acceptance of this examination method. To achieve these goals, and because we were not interested in an evidence synthesis for a single application area, we chose to conduct a scoping review [8]. This decision was guided by the helpful recommendations of the web-based “Right Review” tool and the advice of Munn et al. [9,10].

Accordingly, this scoping review will identify medical applications of passive infrared thermographic imaging in humans for diagnosis, monitoring, prevention, and control or the collection of reference data/normal values. Technical and environmental modalities are to be catalogued and the diagnostic performance of thermography evaluated, the existence of specific reference data reviewed, and finally, research gaps and future tasks will be explored [8].

## 2. Materials and Methods

This scoping review was conducted in accordance with the JBI methodology for scoping reviews [11]. The corresponding protocol was registered in advance in the OSF database (Registration DOI: 10.17605/OSF.IO/TSZ28) and published [8,12]. The presented report of findings follows the Preferred Reporting Items for Systematic Reviews and Meta-Analyses Extension for Scoping Reviews (PRISMA-ScR) [13]. A corresponding PRISMA checklist for scoping reviews can be found in the Appendix A.

### 2.1. Eligibility Criteria and Database Search

The eligibility criteria were identified according to the Population, Concept, Context (PCC) approach and extended with the types of sources and information; details are depicted in Table 1.

PubMed, the Cochrane Library, Embase, and Web of Science were systematically searched using pretested search strategies. Unpublished studies or grey literature were sought in OpenGrey via Data Archiving and Networked Services, OSF, and PROSPERO. In addition, the ‘Published Papers on Thermology or Temperature Measurement’ list provided by Thermology International (http://www.uhlen.at/thermology-international/) (accessed on 17 March 2022) was searched. The time period of the search was from January 2000 to March 2022. Publications before 2000 were not included due to the rapid technical and technological development, as they could contribute little in terms of usable information. Based on the PubMed search strategy shown in Table 2, the searches in the other databases were adapted.

All references obtained were transferred and stored in the RefWorks web-based reference management software package (ProQuest LCC, Ann Arbor, MI, USA), and an initial elimination process of duplicates was conducted program-controlled.

### 2.2. Screening Process and Data Extraction

The titles and abstracts of the retrieved references were screened by two independent researchers according to the inclusion and exclusion criteria detailed above, and further duplicates were removed. Non-matching assignments were discussed and, in cases of doubt, decided upon with a third researcher. Full texts were obtained from the remaining references and these were reviewed by the participating researchers following the same procedure.

Data were extracted into a pre-designed and pre-tested spreadsheet by one researcher, and supervised and reviewed by a second researcher. Data items to be extracted included, among other things, the database source of the reference, the DOI, and further specific data concerning the publication such as author, title, journal, year of publication, etc., information on study design and participants, year of data collection, and finally, decisive information on the investigation like methodology and devices, purpose, intervention, comparator, outcomes, and key findings. Missing technical information on thermographic devices was searched on the internet where possible and mostly taken from technical descriptions of the specific camera models or, in a few cases, from other studies that used the same camera model.

Based on the extracted data, results are divided with regard to the research topics described above (i.e., medical application areas, purpose of application, technical and environmental modalities, diagnostic performance, reference data/normal values) and presented accordingly in graphical, tabular, and narrative form.

## 3. Results

Out of a total of 5027 hits, 72 full texts could be included in the data extraction of the scoping review. Duplicates were automatically removed in refworks (108) and manually excluded (292) in the screening process. Fifty-eight full texts had to be excluded due to the fact that the number of participants was below 25, which was defined as an inclusion/exclusion criteria in the protocol. Twenty-six review articles were identified in the title–abstract scan and the included studies and reference lists were screened accordingly. These reviews each examined one specific use case of thermography, including diabetes, non-contact patient monitoring, breast cancer, systemic sclerosis, ocular surface temperature measurement, atherosclerosis, migraine and headache, thyroid nodules, Raynaud’s phenomenon, temporomandibular dysfunction, perforator mapping, back and neck syndromes, paediatric issues, periprosthetic joint infections, Caesarean section, degenerative joint diseases, muscle injuries, and neurological issues. There was no review that dealt with more than one field of application. The exact numbers of each step are depicted in the PRISMA flow-chart in Figure 1.

A breakdown of the reasons for exclusions in full-text screening with the respective percentages is shown in Figure 2.

### 3.1. Scientific Studies on Medical Thermography: Diseases, Areas of Application, and Primary Study Intentions

To achieve the primary objective of this review, the first step was to create a representative catalogue of all medical use cases of thermography. Ultimately, a total of 38 different indications (close indications have been counted together) have been identified from the extracted data of the included literature sources and are depicted in Table 3.

Thermography is intended to detect (screen), determine (diagnose), observe, and/or evaluate (monitor), among other things, the state of disease and health. A fourth category specific to this work is the assessment of reference values, i.e., normative data. The different categories are visualized with a pie chart in Figure 3.

The use of thermography for indications in specific diseases, as shown in Table 3, ranges from acute appendicitis to wound infection. Figure 4 provides an overview of the distribution in the individual therapeutic disciplines.

The included studies were conducted in the Americas, Asia, and Europe, with India, the USA, China, the UK, and Italy being the most frequently represented countries in descending order. Looking at the included references in terms of their number and year of publication, an increase over time is evident with a peak in 2020/21 with 23 publications. Figure 5 provides an overview of the distribution of publication numbers of the included references summarized in time units of three years.

### 3.2. Thermography Cameras and Technical Properties, Imaging Specifications and Procedures

For the 72 included studies, several thermal imaging camera models from various manufacturers with different performance parameters were used under specified conditions. Also, since some articles described the use of multiple models for comparative analyses, Table 4 lists the information for sixty-four different thermal imagers from fifteen manufacturers and one proprietary system.

Three studies were excluded from this listing, because either information on the manufacturer or model, or on the technical specifications of the device, was missing [20,26,53]. In total, the application of 69 camera models was reported, with FLIR Systems being used in more than half of the investigations. The thermal resolution of all devices ranged from 48 × 47 to 1024 × 678 pixel with a median at 320 × 240 pixel. Thermal sensitivity was between <0.02 °C and 0.5 °C with a median of 0.07 °C. Data for accuracy ranged from ±0.2 °C to ±5 °C with a median of ±2 °C; for eight devices, accuracy was available only as percentages, and for eleven devices no information was given.

Figure 6 provides an overview of the manufacturers of the devices and the respective number of references reporting their application. Since some studies compared several cameras from different manufacturers, the total is more than 72.

The degree to which measurement accuracy depends on environmental conditions, equipment performance parameters, patient management before and during the examination, and procedures during imaging was established quite early on. Appropriate and trained personnel, regular testing, the calibration of cameras, and following basic protocols and standard procedures can contribute to the reliability and reproducibility of the results of the technique. Ring and Ammer described factors influencing medical thermography as early as 2000 and called for adherence to guidelines and standards regarding environment, equipment, patient management, imaging procedures, image processing, and image archiving and reporting [86].

Based on the quality assurance guidelines for clinical thermography presented by the International Academy of Clinical Thermology (IACT) in 2015 [87], we have tabulated the key elements that we consider essential to the imaging process. Table 5 lists the relevant requirements reported in the studies included in our scoping review in graded order. Please note that the respective camera specifications are already provided in Table 4.

### 3.3. Diagnostic Performance

Medical thermography and its ability to provide actionable, clinically applicable information for diagnosis, screening, and monitoring is central to health care, as physician decisions regarding therapeutic or preventive measures rely on the correctness of the results. The most commonly reported measures of the accuracy of a medical test are sensitivity and specificity. Based on these values, the positive predictive value (PPV) and the negative predictive value (NPV) can be determined using prevalence. Another useful quality criterion is the area under the curve (AUC) with respect to the receiver operating characteristic (ROC), which is a graphical plot of the true positive rate (sensitivity) against the false positive rate (1-specificity). The AUC takes a value between 0 and 1, with a higher value indicating a higher diagnostic ability to distinguish between the two conditions in question, typically “diseased” and “not diseased.” Finally, diagnostic accuracy is the proportion of correctly classified true positives and true negatives among all tests results.

Thirty-seven (51%) of the seventy-two included studies evaluated the diagnostic ability of medical thermography in the particular use case. Table 6 contains the corresponding information from the relevant publications.

The data on diagnostic quality were very heterogeneous. Median sensitivity was 87.1 (min 25.0, max 100, *n* = 41), specificity 87.2 (min 50.8, max 99.2, *n* = 40), PPV 72.2 (min 5.7, max 97.4, *n* = 21), NPV 93.3 (min 61.5, max 100, *n* = 22), AUC 86.2 (min 59.0, max 97.1, *n* = 29), and accuracy 89.7 (min 50.2, max 96.3, *n* = 14). Figure 7 visualises the respective data on diagnostic quality in the included studies divided into a time period from 2001 to 2016 and a more recent period from 2017 to 2021.

### 3.4. Thermal Reference Values

Comprehensive normative data of the thermal asymmetry of the human body were published as early as 1988 by Uematsu et al. based on thermographically determined temperature differences (ΔT’s) of 40 paired regions of the body surface of healthy volunteers [90]. In contrast to the absolute temperature, which varies between different individuals, these values of thermal asymmetry obtained from anatomically matching homologous regions are extremely stable and reproducible.

In our scoping review, we identified six publications on normative data of surface temperatures of different body regions, which we will report chronologically. First, Niu et al. measured skin temperature using infrared thermography in 25 body areas in 57 healthy volunteers aged 24 to 80 years in 2001 and compared the respective mean values between left and right, over and under 60 years, and female and male participants [47]. Their results confirm the basic symmetry of the thermoregulatory system in healthy individuals. In 2012, Vardasca et al. studied six regions of upper and lower extremities in 39 healthy male volunteers [48]. Upper arms, forearms, hands, thighs, lower legs, and feet were imaged in anterior and posterior view with an infrared camera and examined with regard to their thermal symmetry. The authors conclude that in the regions of interest under consideration, thermal symmetry does not normally deviate by more than 0.5 °C with a standard deviation of no more than 0.3 °C. Gatt et al. studied thermographic distribution patterns of hand, feet, and lower limbs in 63 healthy male and female adults in 2015 and revealed almost identical symmetry in the thermographic patterns of fingers, palms, shins, and soles [49]. They detected only minimal differences in temperature between the participants and a consistent pattern in toes and fingers. In search of the optimal site for measuring body core temperature and assessing the corresponding bilateral differences with infrared thermal imaging, Vardasca et al. recruited 206 young participants of both sexes in 2019 [50]. Confirming the reliability and reproducibility of their results obtained in relation to the inner canthi of the eye, they emphasize the potential of the related application of infrared imaging for mass screening. In 2020, Matteoli et al. recorded ocular surface temperature in 220 healthy normal participants to obtain reference values stratified by age and sex for a comparison of healthy versus pathologic conditions [51]. In order to be able to use ocular surface temperature for diagnostic purposes, they name some caveats, e.g., age and forehead temperature of the patient in question; however, taking these into account, the present reference values can be applied for the investigation of ocular pathologies. Finally, Lubkowska et al. aimed to thermografically determine the distribution and range of surface temperatures of the entire breast, chest, and abdomen in 105 healthy women with normal weight aged 20–40 years [52]. Despite inter-individual differences, they did not find statistically significant differences in the distribution of temperature between both sides of the body in women with normal weight, confirming basic symmetry in healthy tissue. Based on their 2021 study, they recommend individual interpretation of thermographic results, taking into account the physiological distribution of temperature in both breasts.

## 4. Discussion

### 4.1. Objectives of Research

The purpose of this scoping review was to investigate and catalogue the use of passive thermography in medicine as currently described in the scientific literature. The results show that thermography is used or is being investigated for its applicability in many different medical specialities. From acute appendicitis to wound care, thermographic imaging and temperature measurement are used for diagnostic, screening, and monitoring purposes. In addition, sets of reference data were created that are necessary for differentiating pathological from healthy conditions and are used to understand how pathological processes alter normal surface temperature. Fundamental for the usability of new measurement methods is the assessment of their diagnostic quality, mostly with regard to the so-called gold standard for the corresponding diagnosis. In the following section, the results of these individual sub-objectives of our scoping review will be discussed in more detail.

#### 4.1.1. Use Cases of Passive Infrared Thermography

Medical thermography enables the identification of minimal temperature differences between symmetrical body regions or between the diseased and healthy state of a region. For example, even a slightly pronounced asymmetric temperature pattern between the left and right breast can be an indication of pathologic development [91], while a slightly decreased temperature in relation to a control region may be an early warning sign of pressure injury [68]. Below, we shortly discuss the most recent exemplary studies for each of the identified therapeutic discipline of thermographic use cases.

In the diagnosis of acute appendicitis, as one of the most common indications for emergency surgery, computed tomography is frequently used when medical history, and physical and laboratory examinations are inconclusive, occurring in one-third of suspected cases [14]. The harmful consequences of radiation, especially in children, leads to the search for and evaluation of other diagnostic methods [92]. Here, thermography offers several distinct advantages over conventional methods in terms of portability, invasiveness, complexity of performance, ionizing radiation, and costs [14]. Interestingly, thermography has also been applied in the diagnosis of autism, a disorder of brain development in which one would not necessarily expect temperature changes of the body surface. But by using facial skin temperature to measure emotions, the processing of which is impaired in people with autism spectrum disorder, autistic children can be distinguished from non-autistic children [17]. Breast cancer, as the most common cancer in women, is on the rise worldwide and many countries have initiated massive screening efforts based on the fact that an early detection is associated with lower mortality rates [93,94]. Women with mammographic dense breast tissue are at higher risk for cancer, but at the same time, the sensitivity of mammography is reduced due to the radiological density, so other examination methods are needed in this case [26]. Thermography, approved by the Food and Drug Administration as an additional option for breast cancer screening, offers an easily implementable and cost-effective alternative thanks to new, sophisticated computer-assisted evaluation capabilities [26]. Neck pain is considered one of the most common musculoskeletal disorders with a remarkable economic burden and without one definite treatment [95]. A negative correlation of thermographically measured skin temperature at the myofascial trigger points with electromyographic activity suggests increased muscle activity at rest. Therefore, studies have recommended clinical interventions that promote muscle relaxation and increase blood flow [29]. In the COVID-19 pandemic, it was necessary to conduct rapid screening, especially in critical locations where large numbers of people congregate, to prevent the spread of viruses as much as possible. Here, the use of a portable thermal imaging camera coupled with smartphones to capture thermal images of the back of individuals with and without COVID-19 is reported. A subsequent classification as healthy or diseased was performed using advanced image processing algorithms [30]. Again, thermography is shown to be a tool suitable for low-income countries due to its simplicity of use and low cost. Another promising area for the use of medical thermography is diabetes, particularly type 2, the extent of which is now described as pandemic, with an estimated one in two cases worldwide going undiagnosed [96]. Since middle- and low-income countries show the highest growth rates of type 2 diabetes, simple and cost-effective screening is particularly beneficial in these countries. Thermography of the face and tongue may be used as a basic and non-invasive screening method [33,34]. To better understand the pathophysiology of glaucoma, thermal imaging was used to record ocular surface temperature (OST). The detected differences in OST between glaucomatous and healthy eyes support the assumption of an inflammatory process. Thermography could be used to establish clinical biomarkers for glaucoma, the prognostic value of which needs to be determined in further studies [45]. OST is also under consideration in the next study, which examines its association with systemic risk factors of cardiovascular and ischaemic heart disease. The results show that OST is significantly higher in different ocular regions in individuals with a history of ischemic heart disease. The authors propose to further investigate this correlation to evaluate its potential utility as a clinical screening test [54]. The aim of the next study was to compare a measure of tissue perfusion around the knee, their mottling score, with skin temperature of the same region in patients with septic shock, and their association with survival. Mottling score and skin temperature were not correlated, and neither were they associated with prognosis of day-28 mortality [76]. The usefulness of thermography in assessing postoperative inflammation after third molar removal and its correlation with patient-reported swelling in patients receiving either methylprednisolone or placebo was evaluated in the next research. Two days after surgery, thermograms and swelling were recorded, and temperature difference (ΔT) between the operated and control side was calculated. There was no statistical significant difference observed in ΔT and the correlation with swelling was low. The authors conclude that thermography seems to lack the sensitivity to be able to detect differences in the inflammatory response with regard to methylprednisolone or placebo [66]. A new computer-aided system for assisting diagnosis in rheumatoid arthritis (RA) was tested in women with and without RA. Thermal and RGB images were taken and hand grip strength and gripping force recorded, and an optimal diagnostic model was developed. Interestingly, the RGB delivered better results than the thermal images. However, the images were processed on the basis of their colour, i.e., the temperature distribution in the thermogram was only indirectly taken into account [74]. The number of Caesarean sections is increasing globally, and wound care, especially in women with obesity, is challenging in terms of the prevention of surgical site infections [85]. In a pilot study, thermography was found to improve the prediction of infections and thus may complement subjective assessment in identifying patients with the highest need for antibiotics [85]. This could lead to an improvement in the follow-up of a caesarean section in women with obesity.

In summary, thermography is much more than just an alternative imaging option; above all, it opens up new diagnostic possibilities based on changes in surface temperatures, which were previously not considered in the diagnosis of many diseases. With the capabilities of AI, evaluation options can be improved and facilitated, and diagnostic quality can be increased, a process that is currently emerging in the area of mammography screening, for example [97]. Thermography is still in the early stages of evaluating its potential applications in most areas. Time and again, the included publications emphasized the particular value of thermography for middle- and low-income countries, especially because of its ease of use, portability, and cost-effectiveness.

#### 4.1.2. Technical and Environmental Modalities of Thermographic Imaging, Patient Management

The IACT sets various requirements for a minimum standard of the equipment of the imaging system. In terms of cameras used, we identified some of the most important or most commonly reported for our scoping review. With the exception of three publications, all studies provided sufficient information on the manufacturer and model so that most of the missing technical information could be retrieved from the internet. The use of a specific protocol or recognized standard procedures for thermographic image acquisition was reported in 24 of the included studies. Given the importance attributed to factors such as ambient temperature, patient preparation and acclimatization, and camera orientation and distance in thermographic imaging, this means that two-thirds of the studies either did not follow or at least did not report a protocol in this regard. According to the standards and protocols of the IACT, the minimum thermal resolution of 19,200 pixel was not reached by three devices. The required thermal sensitivity below 0.08 °C was reported for 36 devices and an accuracy of ±2 °C/±2% or better for 44 of 69 devices. A total of twenty-eight models failed to meet the IACT standards, including four due to missing data and seven due to both missing and below standard data.

Given the data from this scoping review, the use of a general standard for thermographic imaging has not yet gained acceptance; in any case, little consideration has been given to the existing IACT standard: Only three of the seventy-two publications included in this scoping review indicated that they followed the IACT standard. For another three and seven studies, respectively, the use of published standard infrared imaging procedures or other comparable standard measures was indicated. A further three studies were conducted using ISO standards for temperature assessment or screening guidelines. Ten study authors reported applying their own protocols. With 48 (67%) of the studies, there was no evidence of the use of a standard regarding the performance of thermographic imaging in more than half of the included references.

Nevertheless, there were notes and reports in 49 publications regarding patient management, which is accurately reported as an important component in the IACT quality assurance guidelines. Careful patient management before and during the examination is considered an important criterion for the accuracy of the images. The training or certification of the personnel assigned to perform thermography, including subsequent analysis, is strongly recommended by the IACT, but there is evidence of the reliance on trained personnel in only eight studies. It is possible that the education and training of personnel, as well as the demands on experience with regard to image analysis, are taking a back seat to the ever-increasing technical capabilities of thermal imaging cameras and the subsequent computer-aided image processing and analysis using machine learning [98]. It is precisely these constantly improving features of thermographic equipment that make them so interesting for use in middle- and low-income countries, as is emphasized several times in the included publications. Specific imaging procedures such as instructions on patient position and posture during imaging, optimal camera distance, and defining regions of interest in advance were reported in varying degrees of detail in 65% of the included studies. Finally, the design and environmental conditions of the room intended for imaging are part of the quality control recommended by the IACT and should meet the thermodynamic properties required for thermal imaging. In this regard, corresponding information was found in 58 (81%) of the included studies.

In summary, since the IACT guidelines were rarely applied and the reports regarding thermographic technology and procedures in the individual articles were very heterogeneous in terms of information and detail, it must be stated that urgent efforts must be made to apply uniform standards, both for thermography itself and for the reporting of related studies.

#### 4.1.3. Diagnostic Performance of Passive Infrared Thermography

Data on diagnostic performance were reported in only 51% of studies. They were heterogeneous, with large ranges of 38 to 92 points between the lowest and the highest values for the individual measures. More recent data showed a nonsignificant trend toward improved performance; for example, the median sensitivity in studies from 2019 was 92.9 compared with 84.1 in studies from years prior. Data on measures of diagnostic performance depend on the choice of threshold, with study authors placing different emphasis on sensitivity and specificity.

Most data were available on breast cancer (*n* = 9) and fever measurements (*n* = 11), with three studies comparing different devices for fever measurement. In some cases, there were noticeable differences between devices, and there was a tendency for improvement in more recent studies. These improvements could be due to technically more enhanced and sophisticated devices but also to advanced methods of image analysis using dedicated software, including artificial intelligence (AI). These results suggest that both breast cancer and fever or body temperature measurements are appropriate areas for future research in thermography.

Breast cancer screening, for which nine studies are available in our scoping review, is a good example of how thermography can be compared with other medical imaging procedures. There are various other screening modalities for breast cancer in particular, with the gold standard currently being X-ray-based mammography. The related sensitivity is reported to be 80%, which drops to 50% or less for young women and women with dense breasts [99]. However, the information on the sensitivity of this gold standard is not uncritical and can certainly be questioned, especially in view of whether it refers to prevalence or incidence screening. In any case, with regard to sensitivity, the mean and standard deviation of the nine studies on thermography included here are 83 ± 23%, with a median of 85%. The high value of the standard deviation is mostly due to a sensitivity of 25% in the earliest study from 2011, which may be regarded as an outlier, as the next highest value is 81.6%. In contrast to mammography, there are no restrictions due to high breast density, which is why thermography is sometimes recommended specifically for this subgroup [100]. Nevertheless, such comparisons must be viewed with caution, as they are based on different approaches and can only serve as a guide at best.

Thermography is not equally suitable for all applications. On the one hand, in 2020, two studies with the computer-aided analysis of thermograms of the face and tongue showed accuracy rates of 89% and 94%, respectively, in discriminating between diseased and healthy participants, and thus could be used as a basic and non-invasive screening method for type 2 diabetes [33,34]. Another successful application was reported in 2021 for the diagnosis of autism, where a sensitivity of 100%, a specificity of 93%, and an accuracy of 96% were achieved by evaluating thermographic images with a customized convolutional neural network (CNN) architecture [17]. On the other hand, in 2017, according to the study authors, thermal imaging was not suitable for the diagnosis of peripheral arterial disease (PAD); however, no data on diagnostic accuracy were available in this study, but the authors identified the fundamental question of whether PAD is associated with significantly reduced temperature, as previously supposed [62]. In 2010, thermography also seemed to be of little help in predicting progression in herpes zoster, as the affected skin areas showed higher temperatures in about half of the patients and lower temperatures in the other half [65]. A randomized controlled trial from 2014 investigated differences in the effect of methylprednisolone versus placebo on postoperative swelling and inflammation after third molar removal, but the authors concluded that thermography was not sufficiently sensitive for detecting differences in inflammatory response [66]. Finally, there are also conflicting assessments by study authors regarding the diagnostic performance of thermography, as the example of temporomandibular dysfunction shows. Two studies from 2013 came to results that were not far apart, with an accuracy of 43.3–50.2% in one study, and 56.3% in the other. Despite this similarity, the authors of the first study concluded that thermography is not sufficiently accurate for diagnosis, while the other authors attested a diagnostic usefulness, which shows that perceived applicability in borderline cases is also in the eye of the beholder. [80,81].

In summary, despite the fact that thermography is not optimal in every setting, positive evaluation prevails. The progressive technical development of the recording devices and the possibilities of applying sophisticated methods of artificial intelligence for the evaluation of the images is shown through the trend of increasing diagnostic performance.

#### 4.1.4. Reference Data on the Thermology of Human Skin

Given the fact that comprehensive normative data on the thermal asymmetry of the human body surface were already published by Uematsu in 1988 [90], there is a need, on the one hand, to verify or update these data by current measurements and, on the other hand, to supplement them with normative data of previously unrecorded areas and regions, e.g., ocular surface temperature. However, the authors of the 1988 study argue that, as expected, absolute temperature can vary over time between non-homologous regions and between individuals, but values of temperature difference (ΔT) collected from anatomically matched homologous regions are small, highly stable, and reproducible [90].

In principle, Niu et al. repeated Uematsu’s study in 2001 with 25, rather than 37, areas and with the further aim of investigating differences between male and female and young and old healthy participants in Taiwan [47]. They found that skin temperatures in 11 areas were slightly lower in the elderly than in the young, for example, in the distal parts of the extremities, and that there were differences between the skin temperatures of elderly women and men in certain body regions. Overall, the temperature differences between the measured areas did not exceed 0.5 °C, broadly confirming the findings of Uematsu et al. Another study, conducted by Verdasca et al. in 2012, confirmed the result that the thermal asymmetry does not exceed the value of 0.5 °C with regard to upper and lower extremities in healthy males [48]. Three years later, Gatt et al. also found no evidence of physiological asymmetries between hands, feet, and lower limbs in 63 healthy male and female adults, and small differences were only found inter-individual [49]. In view of these convincing data on the symmetry of the temperature of contralateral body surfaces, a possibly pathological process cannot be ruled out in the case of measurements deviating from each other, which can, in principle, contribute to finding a diagnosis.

To provide a basis for assessing pathologic processes of the eye, Matteoli et al. collected reference data on ocular surface temperature stratified by age and sex in 2020 [51]. They detected significant differences between ocular regions and dependencies on age and forehead skin temperature as well as on environmental factors. The accurate measurement of eye temperature using thermography provides a better understanding of eye physiology and, in addition to classical diagnostic procedures, offers decision support for therapy. Lubkowska and Chudecka reported reference values for the entire breast surface temperature and assessed the thermal symmetry in healthy women in 2021 [52]. This study also confirms the thermal symmetry of the contralateral body surface, but the differences here can be higher than the 0.5 °C mentioned above. In a pilot study of 1008 women in India, the cutoff was set at 2.5 °C, higher values were considered abnormal, and >3.0 °C was suspected to be breast cancer [21]. Of the 41 women thus identified with ΔT > 3.0 °C, phylloidestumor was detected in one case and breast cancer in 40 women, three of whom had normal mammography. It should be noted that some benign conditions such as infection or inflammation of the breast parenchyma may also alter the temperature distribution and lead to false positive findings [101]. Mammography as the current gold standard for breast cancer screening has several limitations, among which is its unsuitability for women with dense breasts, which especially occurs in younger women, and can lead to false-negative results [102]. Thermographic screening results, in contrast, are not affected by breast density.

Finally, in terms of mass screening to identify people with fever, another study by Vardasca et al. from 2019 provided good results when measuring the temperature of the inner canthi of the eyes with infrared thermography [50]. This result takes on a whole different weight in light of the Corona pandemic, and the demonstrated reliability and reproducibility of this procedure can make a big difference in similar situations.

In summary, our scoping review identified published normative data for several purposes, the first of which was to confirm, in general, the thermal symmetry of the contralateral body regions, indicating possible pathological changes in cases of thermal deviations. Furthermore, the included studies reported reference data of temperature distribution for specific body regions such as the ocular surface and the female breast, and finally determined the best suitable facial region for the most accurate estimation of core body temperature for possible fever mass screening.

### 4.2. Strengths and Limitations

To our knowledge, this is the first scoping review on the application of passive infrared thermography in the medical field. One of the strengths of this work is the adherence to clearly formulated and detailed instructions by experts regarding the JBI methodology used. The protocol-guided execution with predefined objectives and methods guide the knowledge synthesis in a structured and transparent way and minimize the probability of bias. By including diverse sources, a high sensitivity is achieved in the literature search to find as many references as possible to answer the defined research question and thus achieve the research objective in this article.

This review also has some limitations. Due to the very comprehensive research question and the search strategy defined for it, a high number of eligible studies could be identified. However, the limitations to passive thermography and a required group size of at least 25 participants limited the number of studies that were included. The restriction to a group size of at least 25 participants was already specified in the protocol and was based on the expectation of a large number of eligible studies, which would certainly have been associated with a higher workload. It can also be assumed that the additional gain in knowledge would have been comparatively low due to the weaker significance of studies with low statistical power. Thus, it is possible that not all the literature sources that could have been relevant to the aim of this review have been considered and not all application areas, devices, and application modalities, as well as diagnostic performance and reference values for thermography, are included in the results.

### 4.3. Recent Progress in Medical Infrared Thermography and Implications for Further Research

The future prospects and recent progress of infrared thermography are manifold; only a few selected examples will be presented here, for which, there are currently no or only limited satisfactory imaging options. In addition to the contactless, non-invasive, and portable applicability at manageable costs, the rapidly advancing development of infrared cameras on the one hand and artificial intelligence on the other have had a major influence on the growing attractiveness of thermography in recent years. One field in which thermography has been gaining interest and has become one of its most common medical applications, as shown in this review, is the screening of breast cancer. Thermography’s diagnostic quality when screening for breast cancer has improved to a level where it can compete with the gold standard mammography, especially in view of the limited sensitivity of mammography in young women and those with dense breast tissue [103]. The disadvantages of mammography are well known, which is why an important research task for thermography is in the field of breast cancer screening. Thanks to the affordability of 3D sensors and increasing computing power, various methods for generating 3D thermograms have already been developed. New approaches in efficient data handling and storage enable the generation of large-scale 3D thermograms in real-time [104]. These anatomical 3D thermal models can be used, for example, in the prevention, diagnosis, and monitoring of the diabetic foot, but also for gait analysis, with the latter offering a wide range of clinical applications such as the assessment of orthopaedic disabilities, neurological disorders, risk of falls, rehabilitation, and more [105,106,107]. Another promising option is the use of infrared thermography technology for all types of skin lesions, particularly skin cancer but also acne, psoriasis, burns, and other skin conditions, where infrared imaging offers a powerful tool for diagnosis, management, and monitoring [108]. A final example, which also illustrates the variety of potential applications, is the respiratory monitoring of premature babies, which is considered an important indicator in neonatal intensive care. Medical staff usually measure respiratory rates by counting abdominal and chest movements, as alternative adhesive sensors on the skin can be uncomfortable and even painful. A method that integrates non-contact visual and thermal imaging to estimate respiratory rate could be a major advance for this sensitive area [109].

When new measures are introduced into medicine, users vacillate between rejection and uncritical acceptance [110,111]. In the case of thermography, scientific evidence has increased in recent years, opening up many possible fields and modalities of application. However, even in some areas where thermography offers an alternative diagnostic possibility, there may be a possibly unscientific questioning of its goodness due to economic interests. Against this background, it is necessary to evaluate the diagnostic quality in detail, because this is the only way to achieve sufficient evidence for its use. New studies must be designed, conducted, and then reported according to the highest scientific standards—including the application of international standards of thermography procedures. Hence, quite apart from many potential applications not covered in this scoping review, scientific rigour and adherence to protocols and internationally recognized standards are the most important implications for future research.

## 5. Conclusions

Thermography is still in the early stages of evaluating its potential in most medical areas. This scoping review provides comprehensive information on the application of passive infrared thermography in numerous medical specialities where thermographic data are used for diagnostic, monitoring, and screening purposes. Various investigations have shown that the differences between contralateral body surfaces are subtle and therefore deviations can give hints for underlying pathological changes that should be further investigated. Thanks to the constantly growing technical possibilities of thermal imaging cameras and computer-aided image processing and analysis through artificial intelligence, the application is facilitated and the specific training and further education of personnel is simplified. Therefore, in addition to all the applications presented in this scoping review, thermography should be researched more intensively, especially in the area of mass screening and early diagnosis, as it combines the best prerequisites for this, such as portability, non-invasiveness, and automated evaluation options with low resource consumption at reasonable costs. As our results show, thermography is already being used in low- and middle-income countries as an alternative to expensive or hitherto non-existent screening modalities for non-communicable diseases with high prevalence.

## Figures and Tables

**Figure 1 medicina-59-02139-f001:**
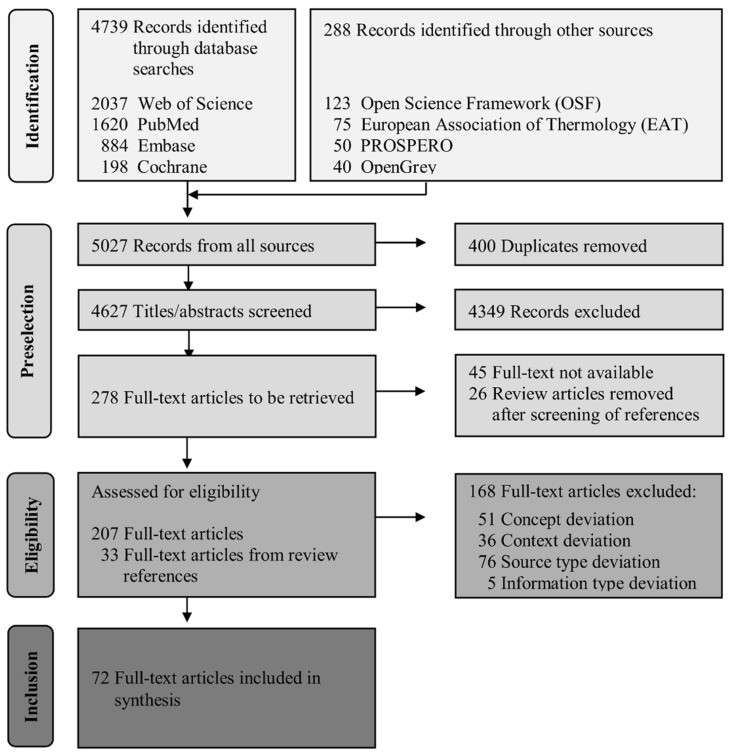
PRISMA flow chart of literature retrieval and selection process according to Tricco et al. [13].

**Figure 2 medicina-59-02139-f002:**
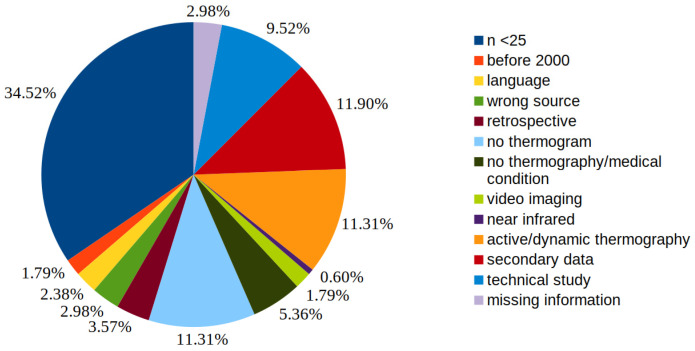
Detailed presentation of the reasons for exclusion in the full text screening.

**Figure 3 medicina-59-02139-f003:**
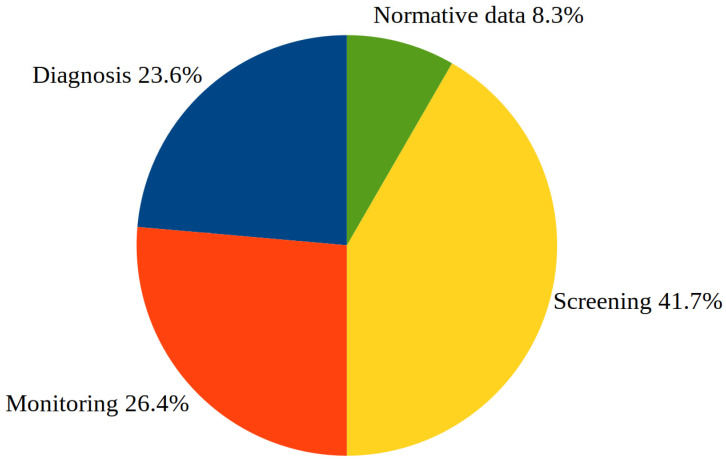
Quantitative assignment of the included studies into categories of the purpose of the respective research.

**Figure 4 medicina-59-02139-f004:**
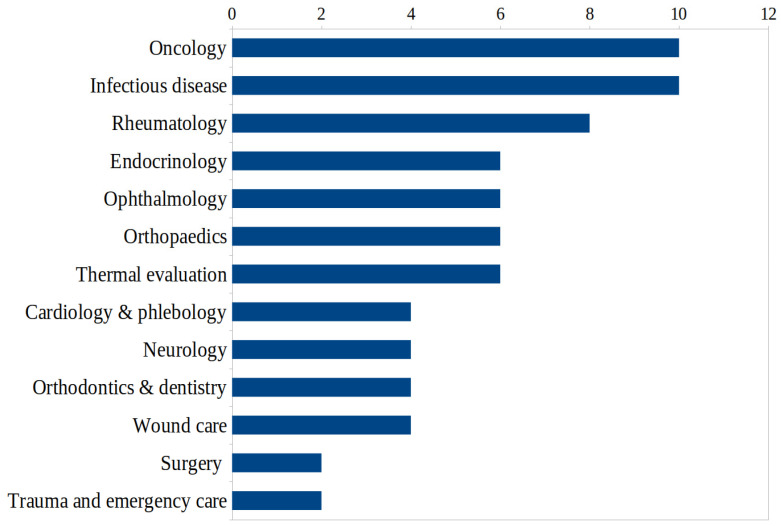
Therapeutic disciplines and respective number of studies with use of thermography identified in this scoping review.

**Figure 5 medicina-59-02139-f005:**
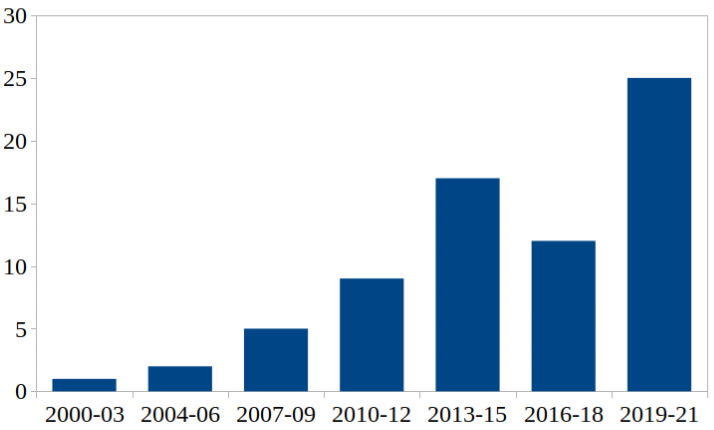
Number of articles included in the scoping review over time using intervals of three years.

**Figure 6 medicina-59-02139-f006:**
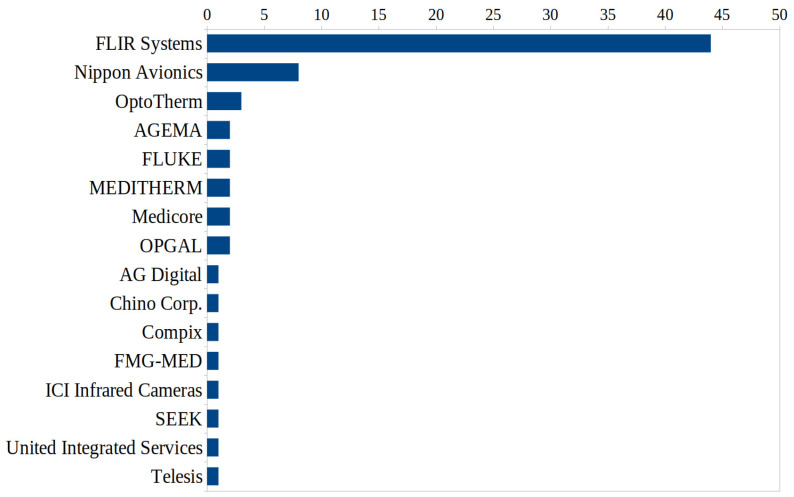
Overview of manufacturers and frequency of use of the respective thermographic devices they produced.

**Figure 7 medicina-59-02139-f007:**
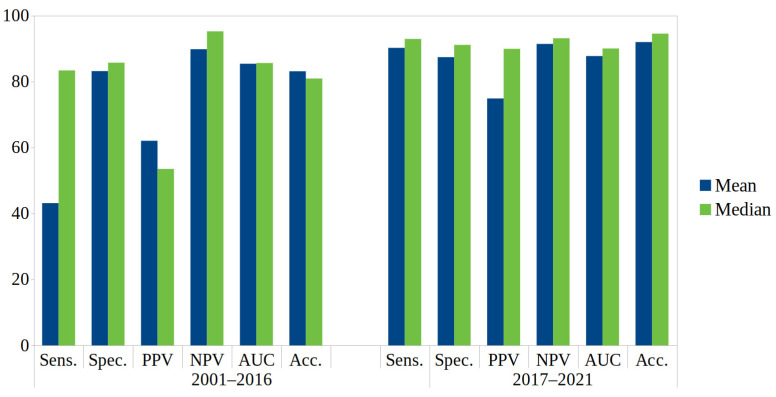
Visualisation of mean and median of diagnostic value characteristic grouped by time periods. The respective numbers (*n* total (n 2001–2016/*n* 2017/2021)) of observations was *n* = 41 (24/17) for sensitivity, *n* = 40 (23/17) for specificity, *n* = 21 (12/9) for PPV, *n* = 22 (13/9) for NPV, *n* = 29 (14/15) for AUC, *n* = 14 (8/6) for accuracy. Sens sensitivity, spec specificity, PPV positive predictive value, NPV negative predictive value, AUC area under the curve, Acc accuracy.

**Table 1 medicina-59-02139-t001:** Structured overview of inclusion and exclusion criteria according to the extended Population, Concept, Context (PCC) approach.

	Inclusion	Exclusion
Population	Human participants (all ages, all sexes)	Animal or in vitro testing
Concept	Application of passive infrared thermographic imaging or temperature measurement for diagnosis (prevention and control), monitoring, or collection of reference data/normal values	Non-medical application, active dynamic thermography, liquid crystal thermography, (functional) near-infrared spectroscopy ((f)NIRS), Fourier infrared spectroscopy (FTIR), diffuse optical imaging (DOS), infrared fundus imaging (no thermography), infrared video recording, infrared tympanic thermometry, dynamic infrared imaging (DIRI)
Context	Clinical, ambulatory, or pre-clinical sites. Any health condition, disease, or medical procedure correlated with the following: changes in blood flow obstruction, destruction, or formation of new blood vessels changes in skin, tissue, or body temperature	Secondary use of thermographic data for analytical or technical optimization
Types of sources	Articles, systematic reviews, meta-analyses, scoping reviews, conference papers, grey literature (theses, dissertations, reports, etc.) reporting quantitative studies with experimental or diagnostic study designs including randomized controlled trials, non-randomized controlled trials, quasi-experimental, case-control studies, and analytical cross-sectional studies Sample size in studies at least 25 per group Any language but title and abstract available in English or German Time period of publication 2000–current	Editorials, viewpoints, opinions, comments, letters, conference abstracts/reports/reviews or summaries, qualitative studies, publications from potential predatory journals and publishers, case reports, case series, retrospective evaluations, technical studies (e.g., image processing, modelling techniques)
Type of information	Basic description of the following: target population site of the examination environmental and technical framework of the assessment disease, anatomical location, objective of the examination, background: preventive or screening, diagnostic, follow-up and monitoring, collection of reference values	Missing information

**Table 2 medicina-59-02139-t002:** Search strategy in PubMed.

Search Step	Search String	Hits
1.	thermography[MeSH Terms] OR spectrophotometry, infrared[MeSH Terms]	81,393
2.	“infrared”[Text Word] AND “temperature”[Text Word]	36,209
3.	“thermology”[Text Word] OR “infrared camera”[Text Word]	1028
4.	(“contact”[Text Word] OR “infrared”[Text Word]) AND thermography[Text Word]	3730
5.	(“therm*”[Text Word] OR “infrared”[Text Word]) AND “imag*”[Text Word]	58,741
6.	diagnosis/prevention and control[MeSH Terms] OR diagnostic techniques and procedures[MeSH Terms:noexp] OR disease progression[MeSH Terms] OR monitoring/physiologic[MeSH Terms] OR mass screening[MeSH Terms:noexp] OR early diagnosis[MeSH Terms]	555,204
7.	reference value[MeSH Terms] OR normal value*[Text Word]	184,175
8.	#1 OR #2 OR #3 OR #4 OR #5	151,614
9.	#6 OR #7	732,765
10.	#8 AND #9	2848
11.	#10 AND (humans[Filter]) AND (2000/1/1:2022/03/17[pdat]) AND (Clinical Study[Filter] OR Comparative Study[Filter] OR Controlled Clinical Trial[Filter] OR Evaluation Study[Filter] OR Journal Article[Filter] OR Pragmatic Clinical Trial[Filter] OR Review[Filter] OR Systematic Review[Filter] OR Technical Report[Filter])	1620

NOTE. * The asterisk is a specific truncation symbol in PubMed.

**Table 3 medicina-59-02139-t003:** Data on use cases of thermography.

Author, Year	Area of Application, Therapeutic Speciality	Country, Number of Participants (Diseased, Control)	Study Intention
Aydemir, U. et al., 2021 [14]	Acute appendicitis, surgery (abdominal)	Turkey, 224 (112, 112)	Diagnosis
Sodi, A. et al., 2014 [15]	Age-related macular degeneration, ophthalmology	Italy, 162 (118, 44)	Monitoring
Murray, B. et al., 2006 [16]	Aphthous ulcers, orthodontics and dentistry	The UK, 52 (26, 26)	Diagnosis
Ganesh, K. et al., 2021 [17]	Autism, neurology	India, 100 (50, 50)	Diagnosis
Kontos, M. et al., 2011 [18]	Breast cancer, oncology (gynaecology)	The UK, 63	Screening
Kolaric, D. et al., 2013 [19]	Breast cancer, oncology (gynaecology)	Croatia, 26	Screening
Yao, X. et al., 2014 [20]	Breast cancer, oncology (gynaecology)	China, 2036	Screening
Rassiwala, M.et al., 2014 [21]	Breast cancer, oncology (gynaecology)	India, 1008	Screening
Zadeh, H.G. et al., 2016 [22]	Breast cancer, oncology (gynaecology)	Iran, 60	Screening
Omranipour, R. et al., 2016 [23]	Breast cancer, oncology (gynaecology)	Iran, 132	Screening
Morales-Cervantes, A. et al., 2018 [24]	Breast cancer, oncology (gynaecology)	Mexico, 206	Screening
Kakileti, S.T. et al., 2020 [25]	Breast cancer, oncology (gynaecology)	India, 470	Screening
Singh, A. et al., 2021 [26]	Breast cancer, oncology (gynaecology)	India, 258	Screening
Carrière, M. et al., 2020 [27]	Burns, wound care	The Netherlands, 32	Monitoring
Wu, C. et al., 2009 [28]	Coccygodynia, orthopaedics	Taiwan, 53	Monitoring
Girasol, C.E. et al., 2018 [29]	Chronic neck pain, orthopaedics	Brazil, 40	Monitoring
Brzezinski, R.Y. et al., 2021 [30]	COVID-19, infectious disease	Israel, 101	Screening
Deng, F. et al., 2015 [31]	Deep venous thrombosis, cardiology (phlebology)	China, 128 (64, 64)	Diagnosis
Sivanandam, S. et al., 2012 [32]	Diabetes type II, endocrinology	India, 62 (30, 32)	Diagnosis
Thirunavukkarasu, U. et al., 2020 [33]	Diabetes type II, endocrinology	India, 160 (80,80)	Diagnosis
Thirunavukkarasu, U. et al., 2020 [34]	Diabetes type II, endocrinology	India, 140 (70, 70)	Diagnosis
Nishide, K. et al., 2009 [35]	Diabetic foot, endocrinology	Japan, 60 (30, 30)	Screening
Zhang, D. 2007 [36]	Facial paresis, neurology	China, 180 (60, 120)	Monitoring
Nguyen, A.V. et al., 2010 [37]	Fever, infectious disease	The USA, 2873	Screening
Hewlett, A.L. et al., 2011 [38]	Fever, infectious disease	The USA, 566	Screening
Selent, M.U. et al., 2013 [39]	Fever, infectious disease	The USA, 855	Screening
Chan, L.S. et al., 2013 [40]	Fever, infectious disease	Hong Kong, 1517	Screening
Ring, E.F.J. et al., 2013 [41]	Fever, infectious disease	Poland, 402	Screening
Sun, G. et al., 2014 [42]	Fever, infectious disease	Japan, 155	Screening
Zhou, Y. et al., 2020 [43]	Fever, infectious disease	USA, 596	Screening
Rabbani, M.J. et al., 2021 [44]	Flap monitoring, surgery (plastic)	Pakistan, 84	Monitoring
Leshno, A. et al., 2022 [45]	Glaucoma, ophthalmology	Israel, 118 (52, 66)	Monitoring
Varju, G. et al., 2004 [46]	Hand osteoarthritis, rheumatology	The USA, 91	Monitoring
Niu, H.H. et al., 2001 [47]	Healthy volunteers, thermal evaluation	Taiwan, 57	Normative data
Vardasca, R. et al., 2012 [48]	Healthy volunteers, thermal evaluation	The UK, 39	Normative data
Gatt, A. et al., 2015 [49]	Healthy volunteers, thermal evaluation	Malta, 63	Normative data
Vardasca, R. et al., 2019 [50]	Healthy volunteers, thermal evaluation	Portugal, 206	Normative data
Matteoli, S. et al., 2020 [51]	Healthy volunteers, thermal evaluation	Italy, 220	Normative data
Lubkowska, A. et al., 2021 [52]	Healthy volunteers thermal evaluation	Poland, 105	Normative data
Matsui, T. et al., 2010 [53]	Influenza, infectious disease	Japan, 92 (57, 35)	Screening
Cohen, G.Y. et al., 2021 [54]	Ischaemic heart disease, cardiology	Israel, 150	Screening
Yamaguchi, M. et al., 2016 [55]	Keratoconjunctivitis sicca, ophthalmology	Japan, 60 (30, 30)	Screening
Tan, L.L. et al., 2016 [56]	Keratoconjunctivitis sicca, ophthalmology	Singapore, 125 (62, 63)	Screening
Zhang, Q. et al., 2021 [57]	Keratoconjunctivitis sicca, ophthalmology	China, 184 (138, 46)	Screening
Kelly-Hope, L.A. et al., 2021 [58]	Lymphatic filariasis, infectious disease	Bangladesh, 153	Monitoring
Su, T. et al., 2017 [59]	Meibomian gland dysfunction, ophthalmology	Taiwan, 154 (89, 65)	Screening
Cohen, E.E.W. et al., 2013 [60]	Mucositis, oncology (otolaryngology)	The USA, 34	Monitoring
Rashmi, R. et al., 2022 [61]	Obesity, endocrinology	India, 150 (50, 50)	Screening
Kyle, D. et al., 2017 [62]	Peripheral arterial disease, cardiology	The UK, 44	Diagnosis
Ilo, A. et al., 2020 [63]	Peripheral arterial disease, cardiology	Finland, 257 (164, 93)	Diagnosis
Romanò, C.L. et al., 2013 [64]	Periprosthetic joint infection, orthopaedics	Italy, 70 (36, 34)	Diagnosis
Han, S.S. et al., 2010 [65]	Post-zoster neuralgia, neurology	Korea, 110	Diagnosis
Christensen, J. et al., 2014 [66]	Postoperative inflammation, orthodontics and dentistry	Denmark, 124 (62, 62)	Monitoring
Cox, J. et al., 2016 [67]	Pressure ulcer, necrosis, wound care	The USA, 67	Monitoring
Cai, F. et al., 2021 [68]	Pressure injury, wound care	China, 349 (82, 267)	Screening
Lasanen, R. et al., 2015 [69]	Rheumatoid arthritis, rheumatology	Finland, 58	Screening
Jones, B. et al., 2018 [70]	Rheumatoid arthritis, rheumatology	Canada, 79 (49, 30)	Monitoring
Umapathy, S. et al., 2019 [71]	Rheumatoid arthritis, rheumatology	India, 60 (30, 30)	Screening
Gatt, A. et al., 2020 [72]	Rheumatoid arthritis, rheumatology	Malta, 83 (32, 51)	Screening
Tan, Y.K. et al., 2020 [73]	Rheumatoid arthritis, rheumatology	Singapore, 37	Monitoring
Alarcón-Paredes, A. et al., 2021 [74]	Rheumatoid arthritis, rheumatology	Mexico, 200 (100, 100)	Screening
Weibel, L. et al., 2007 [75]	Scleroderma, rheumatology	The UK, 41	Monitoring
Ferraris, A. et al., 2018 [76]	Septic shock, trauma and emergency care	France, 46	Monitoring
Park, J.Y. et al., 2007 [77]	Shoulder impingement syndrome, orthopaedics	Korea, 130 (100, 30)	Diagnosis
Sillero-Quintana, M. et al., 2015 [78]	Sports injury, trauma and emergency care	Spain, 201	Diagnosis
Stokholm, J. et al., 2021 [79]	Stroke, neurology	Denmark, 64	Diagnosis
Dibai Filho, A.V.D. et al., 2013 [80]	Temporomandibular disorder, orthodontics	Brazil, 104 (52, 52)	Diagnosis
Woźniak, K. et al., 2015 [81]	Temporomandibular disorder, orthodontics	Poland, 100 (50, 50)	Diagnosis
Damião, C.P. et al., 2021 [82]	Thyroid nodules, endocrinology	Brazil, 113	Diagnosis
Romanò, C.L. et al., 2011 [83]	Total joint arthroplasty, orthopaedics	Italy, 80	Monitoring
Windisch, C. et al., 2016 [84]	Total knee arthroplasty, orthopaedics	Germany, 42	Monitoring
Childs, C. et al., 2019 [85]	Wound infection caesarean section, wound care	The UK, 53	Monitoring

**Table 4 medicina-59-02139-t004:** Manufacturers, camera models with their essential specifications, and clinical conditions for the indication of application, as well as year of publication and respective references.

Manufacturer, City, Country/Region Camera Model	Technical Properties Resolution (Pixel), Thermal Sensitivity, Accuracy (C/%)	Use Case, Year of Publication
AGEMA Infrared Systems, Darmstadt, Germany, AGA Thermovision 900	256 × 240, 0.08 °C at 30 °C, ±1 °C/±1%	Aphthous ulcers, 2006 [16]
AGEMA Infrared Systems, Darmstadt, Germany, AGA Thermovision 782	435 × 435, 0.1 °C at 30 °C	Facial paresis, 2007 [36]
AG Digital Technology Corp., Taipei, Taiwan, ATIR-M301	320 × 240, 0.1 °C, <±1%	Deep venous thrombosis, 2015 [31]
CHINO Corp., Tokyo, Japan, thermopile array	48 × 47, 0.5 °C	Fever, 2014 [42]
Compix Inc., Tualatin, OR, USA, PC200e	244 × 193, 0.1 °C	Hand osteoarthritis [46]
FLIR Systems, Wilsonsville, OR, USA, SC305	320 × 240, <0.05 °C at 30 °C, ±2 °C/±2%	Autism, 2021 [17], Diabetes type II, 2020 [33,34]
FLIR Systems, Wilsonsville, OR, USA, ThermoVision A20	160 × 120, 0.12 °C at 30 °C, ±2 °C/±2%	Breast cancer, 2014 [21] Fever, 2010 [37]
FLIR Systems, Wilsonsville, OR, USA, A315	320 × 240, 0.05 °C, ±1 °C/±1%	Breast cancer, 2020 [25]
FLIR Systems, Wilsonsville, OR, USA, T650sc	640 × 480, 0.02 °C, ±1 °C/±1%	Breast cancer, 2020 [25]
FLIR, Wilsonsville, OR, USA, One Pro	160 × 120, 0.1 °C, ±3 °C/±5%	Burns, 2020 [27], Keratoconjunctivitis, 2021 [57], Pressure injury, 2016 [68]
FLIR Systems, Wilsonsville, OR, USA, T300	320 × 240, 0.05 °C at 30 °C, ±2%	Chronic neck pain, 2018 [29]
FLIR Systems, Wilsonsville, OR, USA, One^®^	160 × 120, 0.1 °C ±3 °C/±5%	COVID-19, 2021 [30], Flap monitoring, 2021 [44]
FLIR Systems, Wilsonsville, OR, USA, ThermaCAM T400	320 × 240, <0.07 °C at 30 °C, ±2 °C/± 2%	Diabetes type II, 2012 [32]
FLIR Systems, Wilsonsville, OR, USA, T360	320 × 240, <0.06 °C at 30 °C, ±2 °C/±2%	Fever, 2013 [39], Temporomandibular disorder, 2013 [80]
FLIR Systems, Wilsonsville, OR, USA, ThermaCAM S40	320 × 240, 0.08 °C, ±2 °C/±2%	Fever, 2013 [40]
FLIR Systems, Wilsonsville, OR, USA, SC640	640 × 480, 0.3 °C at 30 °C, ±2 °C/±2%	Fever, 2013 [41]
FLIR Systems, Wilsonsville, OR, USA, A325sc	320 × 240, <0.05 °C, ±2 °C/±2%	Fever, 2020 [43], Obesity, 2022 [61], Peripheral arterial disease, 2020 [63], Rheumatoid arthritis, 2015 [69]
FLIR, Wilsonsville, OR, USA, ThermoVision A40	320 × 240, <0.1 °C at 30 °C ±2 °C	Healthy volunteers, 2012 [48]
FLIR, Wilsonsville, OR, USA, SC7000	640 × 512 @15μm or 320 × 256 @30μm, <0.03 °C at 30 °C ±1 °C/±1%	Healthy volunteers, 2015 [49]
FLIR Systems, Wilsonsville, OR, USA, E60	320 × 240 <0.05 °C ±2%	Healthy volunteers, 2019 [50]
FLIR Systems, Wilsonsville, OR, USA, A320	320 × 240, <0.05 °C at 30 °C, ±2%	Age-related macular degeneration, 2014 [15], Healthy volunteers, 2020 [51]
FLIR Systems, Wilsonsville, OR, USA, T1030sc	1024 × 678, <0.02 °C, ±1 °C/±1%	Healthy volunteers, 2021 [52]
FLIR Systems, Wilsonsville, OR, USA, C3	128×96, 0.07 °C, ±3 °C/±3%	Lymphatic filariasis, 2021 [58]
FLIR Systems, Wilsonsville, OR, USA, SC300	320 × 240, <0.05 °C, ±2 °C/±2%	Peripheral arterial disease, 2017 [62]
FLIR Systems, Wilsonsville, OR, USA, ThermaCAM E320	320 × 240, 0.08 °C at 30 °C, ±2 °C/±2%	Postoperative inflammation, 2014 [66]
FLIR Systems, Wilsonsville, OR, USA, FLIR i7	140 × 140, <0.1 °C, ±2 °C/±2%	Pressure ulcer, 2016 [67]
FLIR Systems, Wilsonsville, OR, USA, T300	320 × 240, <0.05 °C, ±2%	Rheumatoid arthritis, 2018 [70]
FLIR Systems, Wilsonsville, OR, USA, ThermaCAM T400	320 × 240, <0.07 °C at 30 °C, ±2 °C/±2%	Rheumatoid arthritis, 2019 [71]
FLIR Systems, Wilsonsville, OR, USA, T630	640 × 480, <0.04 °C at 30 °C, ±2 °C/±2%	Rheumatoid arthritis, 2020 [72]
FLIR Systems, Wilsonsville, OR, USA, T640	640 × 480, <0.03 °C at 30 °C, ±2 °C/±2%	Rheumatoid arthritis, 2020 [73]
FLIR Systems, Wilsonsville, OR, USA, ThermaCAM SC500	320 × 240, 0.07 °C, ±2 °C/±2%	Scleroderma, 2007 [75]
FLIR Systems, Wilsonsville, OR, USA, FLIR-E	320 × 240, <0.1 °C, <2%	Septic shock, 2018 [76]
FLIR Systems, Wilsonsville, OR, USA, T335	320 × 240, 0.05 °C, ±2 °C/±2%	Sports injury, 2015 [78]
FLIR Systems, Wilsonsville, OR, USA, T430sc	320 × 240, <0.03 °C, ±2 °C/±2%	Stroke, 2021 [79]
FLIR Systems, Wilsonsville, OR, USA, ThermaCAM SC500	320 × 240, 0.07 °C at 30 °C, ±2 °C/±2%	Temporomandibular disorder, 2015 [81]
FLIR Systems, Wilsonsville, OR, USA, SC620	640 × 480, <0.04 °C, ±2 °C/±2%	Thyroid nodules, 2021 [82]
FLIR Systems, Wilsonsville, OR, USA, FLIR i5	100 × 100, <0,1 °C, ±2 °C/±2%	Total knee arthroplasty, 2016 [84]
FLIR Systems, Wilsonsville, OR, USA, T450sc	320 × 240, <0.03 °C, ±1 °C/±1%	Wound infection caesarean section, 2019 [85]
Fluke, Everett, WA, USA Fluke^®^ Ti9	640 × 480, ≤0.2 °C at 30 °C, ±5 °C/±5%	Acute appendicitis, 2021 [14]
Fluke, Everett, WA, USA, FlexCam Pro^®^	160 × 120, 0.07 °C at 30 °C, ±2 °C/± 2%	Breast cancer, 2018 [24]
FMG-MED, Teheran, Iran, FMG-MED IR	640 × 480, 0.08 °C	Breast cancer, 2016 [23]
ICI, Beaumont, TX, USA 8640 *P*-series	640 × 512 0.03 °C at 30 °C ± 0.2 °C	Fever, 2020 [43]
MEDITHERM, Cheyenne, WY, USA, med2000	320 × 240, <0.1 °C, ±1 °C	Breast cancer, 2011 [18]
MEDITHERM, Cheyenne, WY, USA, IRIS 2000	320 × 240, 0.5 °C, ±1 °C/±1%	Breast cancer, 2020 [25]
Medicore Co., Gyeonggi-do, Korea, IRIS-5000	256 × 240, <0.1 °C	Post-zoster neuralgia, 2010 [65], Shoulder impingement syndrome, 2007 [77]
Nippon Avionics Co., Kanagawa, Japan, NEC Thermo Tracer TH7102WL	320 × 240, 0.07 °C at 30 °C, ±2 °C/±2 %	Breast cancer, 2013 [19]
Nippon Avionics Co., Kanagawa, Japan, InfReC R500	640 × 480, 0.03 °C at 30 °C, ±1 °C	Breast cancer, 2016 [22]
Nippon Avionics, Co., Kanagawa, Japan, NEC Thermotracer TH5108ME	Approx. 750 × 350, 0.1 °C, ±0.7 °C	Diabetic foot, 2009 [35]
Nippon Avionics Co., Kanagawa, Japan, NEC AVIO TVS-2000	256 × 400 0.1 °C at 30 °C	Healthy volunteers, 2001 [47]
Nippon Avionics Co., Kanagawa, Japan, NEC Thermo Tracer TH9420	640 × 480, 0.06 °C	Keratoconjunctivitis, 2016 [56]
Nippon Avionics Co., Kanagawa, Japan, NEC HX0830M1	320 × 240	Keratoconjunctivitis, 2016 [55]
Nippon Avionics Co., Kanagawa, Japan, NEC AVIO ThermoShot F30S	160 × 120, 0.1 °C at 30 °C, ±2 °C/±2%	Periprosthetic joint infection, 2013 [64] Total joint arthroplasty, 2011 [83]
OPGAI, Napels, Italy, Therm-App^®^ (Pro) TH	384 × 288 <0.07 °C ±2 °C/±2%	Glaucoma, 2022 [45] Ischaemic heart disease, 2021 [54]
OptoTherm, Sewickley, PA, USA Thermoscreen	640 × 480, <0.04 °C ±1 °C	Fever, 2010 [37], 2011 [38], 2013 [39]
Proprietary system	256 × 256, 0.06 °C at 30 °C, ±2%	Mucositis, 2013 [60]
SeeK thermal, Santa Barbara, CA, USA Thermal Compact XR	206 × 156, <0.1 °C, ±2 °C/±2%	Rheumatoid arthritis, 2021 [74]
TELESIS, Circleville, OH, USA Telesis Spectrum 9000MB	320 × 240, 0.07 °C	Coccygodynia, 2009 [28]
United Integrated Services Co., Taipei, Taiwan, IT-85	320 × 240, 0.07 °C	Meibomian gland dysfunction, 2017 [59]
Palmer Wahl Instruments, Asheville, NC, USA Fever Alert Imager HSI2000S	Approx. 380 × 415, ~ 0.5 °C at 30 °C, ±2 °C	Fever, 2010 [37]

Note. FLIR Forward Looking Infrared, FMG Fanavaran Madoon Ghermez, ICI Infrared Cameras Incorporation, OPGAI Optical Gas Imaging.

**Table 5 medicina-59-02139-t005:** Specific measures of imaging procedures reported in the included studies.

Standard Protocol	Patient Management	Imaging Modalities	Environmental Control
Standard procedures of the International Academy of Clinical Thermology [87,88] *n* = 3	Instructions for behaviour before imaging (e.g., avoidance of skin lotions, no physical activity) *n* = 25	Trained personnel *n* = 8	Specifications of the examination room (e.g., room without windows, illuminated with neon lights, no sources of thermal energy) *n* = 19
Standard procedures of infrared imaging [86,89] *n* = 3 Others (e.g., [2]) *n* = 7	Instructions for preparation (e.g., take off clothes, remove jewellery) *n* = 36	Instructions of position/posture *n* = 46	Constant/controlled ambient temperature *n* = 48 relative humidity *n* = 22
Standards for temperature assessment, screening guidelines ISO/TR 13154 ISO/TR 80-600 ISO TC121/SC3-IEC SC62D *n* = 3		Pre-specified/fixed camera distance *n* = 47	Ambient temperature/humidity not controlled but recorded *n* = 15
Proprietary protocol/standardized procedure of thermal imaging *n* = 10		Region of Interest (ROI) defined *n* = 47	Acclimatization period *n* = 47
Not reported *n* = 48	Not reported *n* = 23	Not reported *n* = 5	Not reported *n* = 8

NOTE. ISO International organization for standardization, TR Technical report, TC Technical committee, SC Subcommittee, IEC International electrotechnical commission, ROI Region of interest.

**Table 6 medicina-59-02139-t006:** Diagnostic performance of thermography in various application areas and with different evaluation methods.

Use Case, Year of Publication	Study Target	Sensitivity (%), Specificity (%), PPV (%), NPV (%) AUC, Accuracy (%)	Evaluation Method Thermogram, Comparator
Acute appendicitis, 2021 [14]	Diagnosis	77.7 (68.8–85) *, 96.4 (91.1–99) *, 95.6 (89.2–97.3) *, 81.2 (73.3–94.2) * AUC 91.5 (87.1–94.8) *	FLUKE, SmartView^TM^ Desktop Software program v4.3, US, CT, Alvarado score
Autism, 2021 [17]	Diagnosis	Sens. 100, Spec. 93, Acc. 96,	Customised CNN, standard diagnostic procedures
Breast cancer, 2011 [18]	Screening	Sens. 25, Spec. 85, PPV 24, NPV 86	TIFF images classified by 2 physicians, biopsy/surgery
Breast cancer, 2013 [19]	Screening	Sens. 100, Spec. 79	ThermoWeb, ThermoMED version, FNA/surgery
Breast cancer, 2014 [20]	Screening	Sens. 84.4, Spec. 94.0, Acc. 91.7	Automatic software analysis, needle biopsy/surgery
Breast cancer, 2014 [21]	Screening	Sens. 97.6, Spec. 99.2, PPV 83.7, NPV 99.9	Software-aided classification, only those identified in thermographic screening were further clinically examinated
Breast cancer, 2016 [22]	Screening	Sens. 85, NPV 61.5, Acc. 91.9	Detection of thermal asymmetry, biopsy
Breast cancer, 2016 [23]	Screening	Sens. 81.6, Spec. 57.8, PPV 78.9, NPV 61.9, Acc. 69.7	Software-aided classification, histologic results of biopsy/surgery
Breast cancer, 2018 [24]	Screening	Sens. 100, Spec. 68.7, PPV 11.4, NPV 100	Software-aided classification, biopsy
Breast cancer, 2020 [25]	Screening	91.0 (81.8–96) *, 82.4 (78.2–86) *, PPV 50.7, NPV 97.9, AUC 90	AI-based software-aided interpretation, FNA cytology/biopsy
Breast cancer, 2021 [26]	Screening	82.5 (73.2–91.9) *, 80.5 (75.0–86.1) *, 57.8 (47.6–68.0) *, 93.5 (89.7–97.2) *, AUC 84.5, Acc. 81.0 (76.2–85.8) *	AI-based software-aided interpretation, biopsy
Burns, 2020 [27]	Monitoring	Healing potential <14 d vs. ≥14 d Sens. 68, Spec. 95, AUC 89 (83–96) * Healing potential ≤21 d vs. >21 d Sens. 30, Spec. 95, AUC 82 (73–90) *	Software application, Laser Doppler imaging
COVID 19, 2021 [30]	Screening	Sens. 92, Spec 62, PPV 79, NPV 83, AUC 85	Image processing algorithms, COVID-19 PCR test
Diabetes type II, 2012 [32]	Diagnosis	Sens. 90, Spec. 56, PPV 65, NPV 85, AUC 71.1 (58.1–84.2) * Acc. 73	Quick Report software 1.2 by FLIR, HbA1c ≥ 6.5%
Diabetes type II, [33]	Diagnosis	Sens. 88.8, Spec. 91.1, PPV 89.9, NPV 88.9, AUC 94.3, Acc. 89.4	GRAYESS IRT Analyser 6.0, CAD SVM, ADA diagnostic criteria
Diabetes type II, 2020 [34]	Diagnosis	Sens. 92.9, Spec. 95.7, PPV 95.6, NPV 93.1, AUC 80, Acc. 94.3	FLIR software tool, CAD CNN, HbA1c ≥ 6.5%
Fever, 2010 [37]	Screening	OptoTherm 91.0 (85.0–97.0) *, 86.0 (81.0–90.0) *, 17.9 (13.6–22.2) *, 99.6 (99.3–99.8) *, AUC 96.0 FLIR 90.0 (84.0–97.0) *, 80.0 (76.0–84.0) *, 18.4 (13.7–23.0) *, 99.5 (99.1–99.7) *, AUC 92.0 Wahl 80.0 (76.0–85.0) *, 65.0 (61.0–69.0) *, 5.7 (4.1–7.3) *, 99.1 (98.6–99.5) *, AUC 78.2	Skin temperature, oral temperature
Fever, 2011 [38]	Screening	70 (54–83) *, 92 (90–94) *, 42 (31–55) *, 97 (96–99) *, AUC 86.2	ITDS technology, routine protocols
Fever, 2013 [39]	Screening	OptoTherm 83.0 (78–87) *, 86.3 (83–89) *, AUC 92.2 FLIR 83.7 (79–88) *, 85.7 (82–88) *, AUC 92.3 Thermofocus 76.8 (71–82) *, 79.4 (75–83) *, AUC 85.2	ITDS technology, standard protocol, combination of rectal, oral, axillary temperature
Fever, 2013 [40]	Screening	Sens. 57, Spec. 92, PPV 37, NPV 97, AUC 81.2 (76.1–86.3) * at a cut-off of 36.5 °C	ThermaCAM Researcher software, oral or aural temperature
Fever, 2014 [42]	Screening	Sens. 80.5, Spec. 93.3,	Image processing, axillary temperature
Fever, 2020 [43]	Screening	FLIR Sens. 100, Spec. 95 AUC 95 ICI Sens. 100, Spec. 97 AUC 97	Temperatures from thermal images (with BB compensation), oral temperature
Flap monitoring, 2021 [44]	Monitoring	Sens. 98.7, Spec. 75.0, PPV 97.4, NPV 85.7, Acc. 96.3	Temperature gradient colour coding of thermal image, clinical assessment
Glaucoma, 2022 [45]	Monitoring	AUC 69.3	IRT Cronista^®^ 4.0 software, healthy eyes
Influenza, 2010 [53]	Screening	Sens. 88, Spec 89 PPV 93, NPV 82	Software-based screening system, healthy participants
Keratocon-junctivitis sicca, 2016 [55]	Screening	Sens. 83, Spec. 80, AUC 86	Software-aided, Japanese Dry Eye diagnostic criteria
Keratocon-junctivitis sicca, 2016 [56]	Screening	87.1 (76.2–94.3) *, 50.8 (37.9–63.6) *, AUC 72 (63–81) *	OST Analysis V2 software, Schirmer I test
Keratoconjun-ctivitis sicca, 2021 [57]	Screening	Sens. 96, Spec. 91, AUC 79 (73–85) *	FLIR software tool, Japanese Dry Eye diagnostic criteria
Meibomian gland dysfunction, 2017 [59]	Screening	Sens. 90, Spec. 88, AUC 92	Customized computer program, FTBUT, Meibomian gland functional test, Schirmer’s test
Periprosthetic joint infection, 2013 [64]	Diagnosis	89 (74–96) *, 91 (76–98) *, PPV 91 (78–97) *, NPV 88 (74–95) *, Acc. 90	IRTCronista software, standard diagnostic procedures
Pressure injury, 2021 [68]	Screening	Sens. 85.4, Spec. 89.9, PPV 72.2, NPV 95.2, AUC 90 (86–94) *, Acc. 88.8	FLIR One software, Braden scale
Rheumatoid arthritis, 2021 [74]	Screening	Sens. 95.0, Spec. 94.4, AUC 97.1, Acc. 94.7	Computer aided ML, 2010 ACR-EULAR criteria
Scleroderma, 2007 [75]	Monitoring	Sens. 52, Spec. 58, AUC 59 (52–67) *	Thermal difference, clinical diagnosis
Temporoman-dibular dis-order, 2013 [80]	Diagnosis	Sens. 55.8, Spec. 55.8, Acc. 50.2 (38.9–61.5) *	Thermal difference by QuickReport 1.1, RDC/TMD
Temporoman-dibular dis-order, 2013 [81]	Diagnosis	Sens. 46.4, Spec. 95.5, AUC 79.2, Acc. 56.3	Temperature difference, 3-point anamnestic index of TMD
Thyroid nodules, 2021 [82]	Diagnosis	Sens. 96.3, Spec. 99.2, PPV 96.3, NPV 99.2 AUC 96.7 (91.6–100.0) *	Temperature difference curve, FNA biopsy
Wound infection caesaren section, 2019 [85]	Monitoring	Cut-off 33.9 °C: Sens. 92.9, Spec. 36.4, Cut-off 32.65 °C: Sens. 64.3, Spec. 81.8, AUC 75.2 (59.9–90.5) *	Temperature of central abdominal region (umbilicus), wound swabs

NOTE. Sens. sensitivity, Spec. specificity, AUC area under the curve, PPV positive predictive value, NPV negative predictive value, Acc. accuracy, US ultrasound, * 95% confidence interval, CT computed tomography, CNN convolutional neural network, FNA fine-needle aspiration, vs. versus, d days, ADA American Diabetes Association, CAD computer aided diagnosis, SVM support vector machine, ITDS infrared thermal detection system, BB black body, FTBUT fluorescein tear break-up time, ML machine learning, ACR American College of Rheumatology, EULAR European League Against Rheumatism, RDC/TMD research diagnostic criteria for temporomandibular disorders, c-section caesarean section.

## Data Availability

Not applicable.

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
