# Peer review of "Use of Infrared Thermography in Medical Diagnosis, Screening, and Disease Monitoring: A Scoping Review"

_medicina, 2023, doi:10.3390/medicina59122139_

Round 1
Reviewer 1 Report
Comments and Suggestions for Authors
The manuscripts reports the results of a scoping review (SR) on the medical use of passive infrared thermography. The protocol of the SR has been published a priori and the reporting of the SR results follows the PRISMA-ScR guidance which increases transparency of all methodological details of the process of evidence synthesis. The SR has a high methodological standard and its results are comprehensively presented. The manuscript represents solid work that catalogs the use of passive thermography in medical applications and comprehensively discusses its utility in an understandable manner.
Minor remarks:
- The authors decided to limit their search only to studies with a sample size of at least 25 per group. This inclusion criterion seems a little bit arbitrary and should be explained. Figure 1 shows that 76 full texts were excluded from the SR due to source type deviation which includes - but is not limited to - the sample size requirement. How many studies had to be excluded due to the sample size restriction? This information should be also be incorporated in the limitations paragraph in the Discussion where the authors quite vaguely write that this eligibilty criterion "limit the number of studies to be included" (l. 577).
- When giving an overall summary of the data on diagnostic utility (sensitivity, specificity, PPV, NPV, AUC, and accuracy) in line 260ff, the authors should report minimum and maximum (in brackets) instead of just giving the range.
l. 89-91: The paragraph is a left-over from the template file which should be eliminated.
Author Response
We would like to thank the reviewers for their kind review of our comprehensive manuscript and for their valuable comments. We have revised our manuscript and hope to meet all requirements.
Reviewer 1: Comments and Suggestions for Authors
The manuscripts reports the results of a scoping review (SR) on the medical use of passive infrared thermography. The protocol of the SR has been published a priori and the reporting of the SR results follows the PRISMA-ScR guidance which increases transparency of all methodological details of the process of evidence synthesis. The SR has a high methodological standard and its results are comprehensively presented. The manuscript represents solid work that catalogs the use of passive thermography in medical applications and comprehensively discusses its utility in an understandable manner.
Minor remarks:
- The authors decided to limit their search only to studies with a sample size of at least 25 per group. This inclusion criterion seems a little bit arbitrary and should be explained.
This objection is certainly justified and we are grateful for the opportunity to improve our manuscript in this respect. This restriction was already published in the protocol of our review: "Given the expected volume of literature and the independence of the outcomes of this scoping review from the size of the included studies, a minimum sample size of 25 individuals per group studied was established." On the one hand, this was a proposal of one of the reviewers of our protocol and, on the other hand, we thought it made sense, as we expected a very large number of studies. In addition, a larger number of subjects or patients also ensures greater statistical power and thus contributes to study quality.
Changes made in the manuscript:
4.2 Strengths and limitations, page 27, line 591ff:
The restriction to a group size of at least 25 participants was already specified in the protocol and was based on the expectation of a large number of eligible studies, which would certainly have been associated with a higher workload. It can also be assumed that the additional gain in knowledge would have been comparatively low due to the weaker significance of studies with low statistical power.
Figure 1 shows that 76 full texts were excluded from the SR due to source type deviation which includes - but is not limited to - the sample size requirement. How many studies had to be excluded due to the sample size restriction? This information should be also be incorporated in the limitations paragraph in the Discussion where the authors quite vaguely write that this eligibilty criterion "limit the number of studies to be included" (l. 577).
In any case, it makes sense to communicate this figure openly, which is why we have now included this information centrally in the "Results" section of our manuscript.
Changes made in the manuscript:
3. Results, page 5, line 137ff:
Duplicates were automatically removed in refworks (108) and manually excluded (292) in the screening process. Fifty-eight full texts had to be excluded due to the fact that the number of participants was below 25, which was defined as an inclusion/exclusion criteria in the protocol.
- When giving an overall summary of the data on diagnostic utility (sensitivity, specificity, PPV, NPV, AUC, and accuracy) in line 260ff, the authors should report minimum and maximum (in brackets) instead of just giving the range.
Thank you for this suggestion, we have added the data accordingly.
Changes made in the manuscript:
3.3 Diagnostic performance, page 21, line 254ff:
Median sensitivity was 87.1 (min 25.0, max 100, n=41), specificity 87.2 (min 50.8, max 99.2, n=40), PPV 72.2 (min 5.7, max 97.4, n=21), NPV 93.3 (min 61.5, max 100, n=22), AUC 86.2 (min 59.0, max 97.1, n=29), and accuracy 89.7 (min 50.2, max 96.3, n=14).
l. 89-91: The paragraph is a left-over from the template file which should be eliminated.
Thank you very much, we all overlooked that.

Reviewer 2 Report
Comments and Suggestions for Authors
The presented manuscript, titled "Use of Infrared Thermography in Medical Diagnosis, Screening, and Disease Monitoring: A Scoping Review," is concise, clear, and lays the groundwork for understanding the importance of medical thermography in clinical decision-making. The text adheres to PRISMA-ScR principles, providing comprehensible details about the review process alongside the discussion of the role of medical thermography in healthcare.
While the overall quality of the text and the review process is commendable, a few minor issues should be addressed:
1. Some information about the excluded studies should be provided, preferably in graphical form such as Fig.2 or another summarizing method. Concerns arise due to the very small fraction of selected studies compared to the 1620 obtained in the 11th search strategy. Potential reasons, such as a small number of patients involved or purely technical data, should be clarified.
2. The text mentions the identification of 26 reviews, and it would be valuable to include information about their purpose and how they differ from the presented review.
3. Discrepancies between the data in Fig.1 and Table 1 need clarification. Identifying and addressing the reason for these discrepancies is essential for maintaining accuracy and reliability.
4. The data from Table 6 should be presented additionally in graphical form and compared with other imaging modalities. Similarly, a summary of the data from Table 4 would provide additional value.
5. In lines 509-511, if the broad mapping was deemed necessary, the statement appears to negate claims about the usefulness of the technique. It is suggested to either remove the statement altogether or provide concrete evidence for such claims, avoiding groundless speculation.
6. As a final note, the review sometimes gives the impression of being self-referential. In a medical paper, this might be less justified unless it serves an educational/methodological purpose (how to write an scoping review). Clarification or justification for this approach, if intentional, should be provided in the introduction and reflected in the overall manuscript and abstract.
Overall, the text is well-written, and the review is well-performed, but addressing these minor issues would enhance the overall clarity and completeness of the manuscript.
Author Response
We would like to thank the reviewers for their kind review of our comprehensive manuscript and for their valuable comments. We have revised our manuscript and hope to meet all requirements.
Reviewer 2: Comments and Suggestions for Authors
The presented manuscript, titled "Use of Infrared Thermography in Medical Diagnosis, Screening, and Disease Monitoring: A Scoping Review," is concise, clear, and lays the groundwork for
understanding the importance of medical thermography in clinical decision-making. The text adheres to PRISMA-ScR principles, providing comprehensible details about the review process
alongside the discussion of the role of medical thermography in healthcare.
While the overall quality of the text and the review process is commendable, a few minor issues should be addressed:
1. Some information about the excluded studies should be provided, preferably in graphical form such as Fig.2 or another summarizing method. Concerns arise due to the very small fraction of selected studies compared to the 1620 obtained in the 11th search strategy. Potential reasons, such as a small number of patients involved or purely technical data, should be clarified.
Unfortunately, it is not possible for us to provide a list of the reasons for 4349 articles excluded in the title/abstract scan, although it turned out that most of the exclusions were actually already based on the title, which often contained a reference to "infrared" and/or "thermography", as intended in the systematic search, but then provided no further evidence for inclusion. With the search for text words, as can be seen in the PubMed example, the search strategy is of course also very broad, so the large number of exclusions is not very surprising. However, in an effort to be more precise in the second step, we have listed the exclusions from the full-text screening in detail and displayed them graphically.
Changes made in the manuscript:
3. Results, page 6, line 153ff:
A breakdown of the reasons for exclusions in full-text screening with the respective percentages is shown in Figure 2.
Figure 2. Detailed presentation of the reasons for exclusion in the full text screening.
2. The text mentions the identification of 26 reviews, and it would be valuable to include information about their purpose and how they differ from the presented review.
That is indeed an interesting question, and we have included the answer in our manuscript. As you can see, all reviews have dealt with a specific use case of thermography within the medical field of application.
Changes made in the manuscript:
3. Results, page 5, line 140ff:
Twenty-six review articles were identified in the title-abstract scan and the included studies and reference lists were screened accordingly. These reviews each examined one specific use case of thermography, including diabetes, non-contact patient monitoring, breast cancer, systemic sclerosis, ocular surface temperature measurement, atherosclerosis, migraine and headache, thyroid nodules, Raynaud's phenomenon, temporomandibular dysfunction, perforator mapping, back and neck syndromes, paediatric issues, periprosthetic joint infections, Caesarean section, degenerative joint diseases, muscle injuries, and neurological issues. There was no review that dealt with more than one field of application.
3. Discrepancies between the data in Fig.1 and Table 1 need clarification. Identifying and addressing the reason for these discrepancies is essential for maintaining accuracy and reliability.
Thank you for the careful checking! In fact, the flowchart shows an outdated and therefore incorrect overview, we have corrected this with the figures from the final literature research from March 2022.
4. The data from Table 6 should be presented additionally in graphical form and compared with other imaging modalities. Similarly, a summary of the data from Table 4 would provide additional value.
Following your comment, we have added a graphical representation to Table 6. In the Discussion, we added a paragraph to exemplarily compare the sensitivity of thermography in the detection of breast cancer with the gold standard. It would undoubtedly be highly interesting and important to compare the accuracy of thermography with other imaging techniques in different use cases, but this is beyond the bounds of this scoping review, though it would be an exciting topic for a further publication. We decided against a graphical representation for Table 4 because we found it difficult to visualize the data. However, a summary of the data with the most important information is provided directly below the table in narrative form.
Changes made in the manuscript:
3.3 Diagnostic performance, page 21, line 258ff:
Figure 7 visualises the respective data on diagnostic quality in the included studies divided into a time period from 2001-2016 and a more recent period from 2017-2021.
Figure 7. Visualisation of mean and median of diagnostic value characteristic grouped by time periods. The respective numbers (n total(n 2001-2016/n 2017/2021)) of observations was n=41(24/17) for sensitivity, n=40(23/17) for specificity, n=21(12/9) for PPV, n=22(13/9) for NPV, n=29(14/15) for AUC, n=14(8/6) for accuracy. Sens sensitivity, spec specificity, PPV positive predictive value, NPV negative predictive value, AUC area under the curve, Acc accuracy.
4.1.3. Diagnostic performance of passive infrared thermography, page 25, line 475ff:
Breast cancer screening, for which nine studies are available in our scoping review, is a good example of how thermography can be compared with other medical imaging procedures. There are various other screening modalities for breast cancer in particular, with the gold standard currently being X-ray-based mammography. The related sensitivity is reported to be 80%, which drops to 50% or less for young women and women with dense breasts [99]. However, the information on the sensitivity of this gold standard is not uncritical and can certainly be questioned, especially in view of whether it refers to prevalence or incidence screening. In any case, with regard to sensitivity, the mean and standard deviation of the nine studies on thermography included here are 83±23%, with a median of 85%. The high value of the standard deviation is mostly due to a sensitivity of 25% in the earliest study from 2011, which may be regarded as an outlier, as the next higher value is 81.6%. In contrast to mammography, there are no restrictions due to high breast density, which is why thermography is sometimes recommended specifically for this subgroup [100]. Nevertheless, such comparisons must be viewed with caution, as they are based on different approaches and can only serve as a guide at best.
5. In lines 509-511, if the broad mapping was deemed necessary, the statement appears to negate claims about the usefulness of the technique. It is suggested to either remove the statement altogether or provide concrete evidence for such claims, avoiding groundless speculation.
We have decided to remove this statement as it actually contradicts the broad applicability of thermography to a certain extent.
6. As a final note, the review sometimes gives the impression of being self-referential. In a medical paper, this might be less justified unless it serves an educational/methodological purpose (how to write an scoping review). Clarification or justification for this approach, if intentional, should be provided in the introduction and reflected in the overall manuscript and abstract.
We are not entirely sure whether we understand this comment correctly, but self-referentiality was not a deliberate intention.
If you are referring to the fact that we often quote from the included studies, then this is indeed to some extent intentional, as these were "validated" to some extent by the strict inclusion and exclusion criteria. However, we hope that our revision and the inclusion of further literature have somewhat relativized this impression.
Overall, the text is well-written, and the review is well-performed, but addressing these minor issues would enhance the overall clarity and completeness of the manuscript.
Thank you for your positive evaluation!
Reviewer 3 Report
Comments and Suggestions for Authors
The article is devoted to the use of infrared thermography in medical diagnostics and screening.
The article is descriptive.
Suggestions:
. Authors need to identify criteria and compare infrared thermography datasets with reference to literary sources.
2. The authors need to compare modern means of infrared thermography with reference to literary sources.
3. In the article, it is advisable to reduce the size of tables 3, 4, 6. It is necessary to highlight the main sources that determine modern trends in infrared thermography research.
4. Figure 5 is not very informative.
5. In the conclusions based on the analysis of many articles, it is necessary to highlight promising directions of research in infrared thermography.
Author Response
We would like to thank the reviewers for their kind review of our comprehensive manuscript and for their valuable comments. We have revised our manuscript and hope to meet all requirements.
Reviewer 3: Comments and Suggestions for Authors
The article is devoted to the use of infrared thermography in medical diagnostics and screening.
The article is descriptive.
Suggestions:
1. Authors need to identify criteria and compare infrared thermography datasets with reference to literary sources.
Could you please explain this requirement in more detail, as we do not fully understand what exactly you mean by this. Do you mean publicly available datasets that contain thermograms? What should the criteria be? We have set clear content criteria for the studies to be included in this review, do you think these are incomplete? We would be very grateful for further information and would be happy to implement your suggestions.
2. The authors need to compare modern means of infrared thermography with reference to literary sources.
We welcome this proposal to add a paragraph to our discussion on this point.
Changes made in the manuscript:
4.3 Recent progress in medical infrared thermography and implications for further research, page 27, line 600ff:
The future prospects and recent progress of infrared thermography are manifold; only a few selected examples will be presented here, for which there are currently no or only limited satisfactory imaging options. In addition to the contactless, non-invasive and portable applicability at manageable costs, the rapidly advancing development of infrared cameras on the one hand and artificial intelligence on the other have had a major influence on the growing attractiveness of thermography in recent years. One field in which thermography has been gaining interest and has become one of its most common medical application, as shown in this review, is the screening of breast cancer. Thermography’s diagnostic quality when screening for breast cancer has improved to a level where it can compete with the gold standard mammography, especially in view of the limited sensitivity of mammography in young women and those with dense breast tissue [103]. The disadvantages of mammography are well known, which is why an important research task for thermography is in the field of breast cancer screening. Thanks to the affordability of 3D sensors and increasing computing power, various methods for generating 3D thermograms have already been developed. New approaches in efficient data handling and storage enable the generation of large-scale 3D thermograms in real-time [104]. These anatomical 3D thermal models can be used, for example, in the prevention, diagnosis and monitoring of the diabetic foot, but also for gait analysis, with the latter offering a wide range of clinical applications such as the assessment of orthopaedic disabilities, neurological disorders, risk of falls, rehabilitation and more [105-107]. Another promising option is the use of infrared thermography technology for all types of skin lesions, particularly skin cancer, but also acne, psoriasis, burns and other skin conditions, where infrared imaging offers a powerful tool for diagnosis, management and monitoring [108]. A final example, which also illustrates the variety of potential applications, is respiratory monitoring of premature babies, which is considered an important indicator in neonatal intensive care. Medical staff usually measure respiratory rate by counting abdominal and chest movements, as alternative adhesive sensors on the skin can be uncomfortable and even painful. A method that integrates non-contact visual and thermal imaging to estimate respiratory rate could be a major advance for this sensitive area [109].
3. In the article, it is advisable to reduce the size of tables 3, 4, 6. It is necessary to highlight the main sources that determine modern trends in infrared thermography research.
Thank you for your assessment that we can understand, but we would like to leave it to the editors to decide whether tables need to be reduces in size, as these tables are reporting and representing the central results of the review.
Modern trends in infrared thermography research have been added to 4.3 “Recent progress in medical infrared thermography and implications for further research” as depicted above.
4. Figure 5 is not very informative.
Thank you for your assessment. In this case, again, we would like to leave it to the editors to decide whether the figure should be removed, as the other reviewers did not make any comments on this.
5. In the conclusions based on the analysis of many articles, it is necessary to highlight promising directions of research in infrared thermography.
Following your suggestion, we have added the following sentences to the conclusion. Further promising directions have already been discussed in the amendments to point 2 of your report.
Changes in the manuscript:
5. Conclusions, page 28, line 655ff:
in addition to all the applications presented in this scoping review, thermography should be researched more intensively, especially in the area of mass screening and early diagnosis, as it combines the best prerequisites for this, such as portability, non-invasiveness, automated evaluation options with low resource consumption at reasonable costs.